# The T2 Toxin Produced by *Fusarium* spp. Impacts Porcine Duodenal Nitric Oxide Synthase (nNOS)-Positive Nervous Structures—The Preliminary Study

**DOI:** 10.3390/ijms21145118

**Published:** 2020-07-20

**Authors:** Andrzej Rychlik, Slawomir Gonkowski, Ewa Kaczmar, Kazimierz Obremski, Jaroslaw Calka, Krystyna Makowska

**Affiliations:** 1Department of Clinical Diagnostics, Faculty of Veterinary Medicine, University of Warmia and Mazury in Olsztyn, Oczapowskiego 14, 10-957 Olsztyn, Poland; andrzej.rychlik@uwm.edu.pl (A.R.); ewa.kaczmar@uwm.edu.pl (E.K.); 2Department of Clinical Physiology, Faculty of Veterinary Medicine, University of Warmia and Mazury in Olsztyn, Oczapowskiego 13, 10-957 Olsztyn, Poland; slawomir.gonkowski@uwm.edu.pl (S.G.); jaroslaw.calka@uwm.edu.pl (J.C.); 3Department of Veterinary Prevention and Feed Hygiene, Faculty of Veterinary Medicine, University of Warmia and Mazury in Olsztyn, Oczapowskiego Str. 13, 10-718 Olsztyn, Poland; kazimierz.obremski@uwm.edu.pl

**Keywords:** nitric oxide, nitric oxide synthase, nNOS, enteric nervous system, porcine, mycotoxins, T2 toxin

## Abstract

T2 toxin synthetized by *Fusarium* spp. negatively affects various internal organs and systems, including the digestive tract and the immune, endocrine, and nervous systems. However, knowledge about the effects of T2 on the enteric nervous system (ENS) is still incomplete. Therefore, during the present experiment, the influence of T2 toxin with a dose of 12 µg/kg body weight (b.w.)/per day on the number of enteric nervous structures immunoreactive to neuronal isoform nitric oxide synthase (nNOS—used here as a marker of nitrergic neurons) in the porcine duodenum was studied using the double immunofluorescence method. Under physiological conditions, nNOS-positive neurons amounted to 38.28 ± 1.147%, 38.39 ± 1.244%, and 35.34 ± 1.151 of all enteric neurons in the myenteric (MP), outer submucous (OSP), and inner submucous (ISP) plexuses, respectively. After administration of T2 toxin, an increase in the number of these neurons was observed in all types of the enteric plexuses and nNOS-positive cells reached 46.20 ± 1.453% in the MP, 45.39 ± 0.488% in the OSP, and 44.07 ± 0.308% in the ISP. However, in the present study, the influence of T2 toxin on the intramucosal and intramuscular nNOS-positive nerves was not observed. The results obtained in the present study indicate that even low doses of T2 toxin are not neutral for living organisms because they may change the neurochemical characterization of the enteric neurons.

## 1. Introduction

Mycotoxins are secondary metabolites of various fungi species. The main producers of these harmful substances are *Penicillium*, *Aspergillus,* and *Fusarium* spp. [1]. The latter genus mostly synthetizes toxins called trichothecenes which, due to differences in their chemical structure, are divided into A, B, C, or D types. From among these four types, A and B trichothecenes show the strongest toxicity and therefore they are the greatest threat to the health and life of humans and animals [2]. Type A trichothecenes includes, among others, T2 toxin (T2), HT2 toxin, diacetoxyscripenol, and neosolaniol—among which T2, commonly occurring in food and feed, is characterized by the highest toxicity [2,3].

According to current knowledge, T2 is recognized as a mycotoxin, which causes a broad spectrum of adverse effects on the living organism. Previous studies have shown that after exposure to the T2 toxin, the most intense changes appear in the digestive tract, as well as in the immune, endocrine, and nervous systems. Moreover, it has caused toxicosis outbreaks in many countries, such as Japan, USA, China, and Canada [4,5]. However, although it is known that T2 intoxication results in a number of gastrointestinal symptoms, it also causes damage within the nervous system, and knowledge about the effects of T2 on the enteric nervous system (ENS) is still incomplete [6,7].

Due to the presence of millions of neurons dispersed in the wall of the gastrointestinal (GI) tract, the ENS is characterized by a high degree of functional autonomy [7,8]. Enteric neurons manage the regulation of the majority of GI tract activities, such as intestinal motility, secretion of the digestive enzymes, absorption of nutrients, intestinal blood flow, and many others [8,9].

Enteric nerve cells form intramural plexuses, whose location in the gastric and intestinal wall depends on the segment of the GI tract and mammalian species. In the small intestine of large mammals (including humans and pigs) there are three types of ganglionated intramural plexuses. In the smooth muscular layer, between the longitudinal and circular fibers, is the myenteric plexus (MP). In turn, within the submucous layer, two types of plexuses are distinguished. The first of them is the outer submucous plexus (OSP), which lies close to the internal side of the circular fibers of the intestinal muscular coat; and the second is the inner submucous plexus (ISP), which is situated right next to the lamina propria of the mucosa [10,11,12]. Such organization of the enteric plexuses makes the porcine ENS similar to the human enteric nervous system, but it should be underlined that, besides the building and distribution of intestinal plexuses, the human and porcine enteric neurons are also similar in their essential, neurochemical, physiological, and biochemical properties. For this reason, the domestic pig is considered to be a good animal model for studies on the pathological processes occurring in the human body influenced by the ENS [9,13], all the more so since it is known that the enteric neurons can undergo structural, functional, or chemical changes as a result of various pathological stimuli [6,9,10,14,15].

Complicated and multidirectional ENS functions are possible thanks to several dozen biologically active substances produced by enteric neurons. To date—apart from acetylcholine, which is the main ENS neurotransmitter—several other active substances have been described in the nervous structures within the GI tract [6,8,9,10,14,16,17]. One of them is nitric oxide (NO), which unlike the majority of neuronal factors in the ENS, is a gaseous neurotransmitter.

During studies on the nervous system (including ENS), a neuronal isoform of nitric oxide synthase (nNOS), catalyzing the production of NO from L-arginine, is often used as a marker of nitrergic nervous structures [9,18]. This is because NO is oxidized to nitrite and nitrate in a matter of seconds after its synthesis. According to the current knowledge, nNOS-positive nervous structures are present in the ENS of numerous species, including humans, and the number of nitrergic neurons depends on the animal species and the studied segment of the GI tract [19,20,21,22,23]. In the ENS, nitric oxide is classified as one of the most important inhibitory factors. Previous studies have reported that NO causes relaxation of the gastrointestinal smooth muscles and inhibits the secretion of electrolytes and intestinal hormones [24]. Furthermore, since nitric oxide is an important vasodilator, it is involved in the regulation of mesenteric and intestinal blood flow [9,21].

However, many aspects of NO functions in intestinal neurons are not clear. Among others, the participation of nitrergic enteric neuronal cells in pathological processes within the intestine is not fully known. Some previous reports have suggested that NO has neuroprotective functions in the ENS and participates in pathological processes in the intestine, however, knowledge of this topic is still rather scarce [9,22,25].

Therefore, the present investigation studied the influence of T2 toxin on the nNOS-positive nervous structures in the porcine duodenum and, due to the abovementioned high similarities in the organization and functioning of the ENS between humans and pigs, the results obtained during this experiment will help to understand the mechanisms of the impact of T2 on the human digestive tract.

## 2. Results

In the present investigation, nervous structures immunopositive to nNOS were observed in all types of enteric plexuses of both studied groups of animals.

Under physiological conditions, the number of nNOS-like immunoreactive (nNOS-LI) neurons was relatively high in all types of plexuses studied. It reached up to 38.28 ± 1.147%, 38.39 ± 1.244%, and 35.34 ± 1.151% of all neuronal cells labeled with PGP 9.5 in the MP (Figure 1), OSP (Figure 2), and ISP (Figure 3), respectively. For nerve fibers, nNOS-positive structures were not so abundant. The nNOS-LI nerves observed in the wall of the duodenum were generally slender and short. In the mucosal layer, the average number of fibers immunoreactive to nNOS amounted to 9.66 ± 0.62 nerves per observation field; whereas in the muscular layer, such nerves were more numerous and their average number reached 24.73 ± 1.008.

During the present study, the influence of T2 toxin on the nitrergic nervous structures was noted in the majority of the studied ENS elements. In all types of the enteric plexuses, the observed changes were noticeable and statistically significant. In contrast to neuronal cells, T2-induced fluctuations in the density of intramucosal, intramuscular, and intraganglionic nerve fibers were not detected.

Among all types of the enteric plexuses, the greatest differences between control pigs and animals treated with T2 were observed within the ISP, where the studied mycotoxin increased the percentage of nNOS-LI perikarya to 44.07 ± 0.308% (by almost 9 percentage points—pp). In the OSP after T2 toxin administration, the amount of nNOS-LI cells was 45.39 ± 0.488% and it was higher than in the control animals by about 7 pp. Changes between T2 and C groups were observed in the MP as well. In this case, the percentage of neurons immunopositive to nNOS amounted to 46.20 ± 1.453%, and it was higher by almost 8 pp in reference to the physiological state (Table 1).

As mentioned above, contrary to the enteric neurons, T2 toxin administration did not affect the density of intramucosal, intramuscular, or intraganglionic nNOS-LI nerve fibers in the porcine duodenum, and the slight differences in the number of such nerves noted in the mucosal and muscular layers between both groups of animals were not statistically significant (Table 1). Moreover, T2 toxin did not change the morphology of nitrergic nerves. In animals treated with the mycotoxin, such nerves (like in the control animals) were rather delicate and thin.

During the present investigation, an average surface area of enetric ganglia was also studied (Table 2). However, the differences between T2 and C groups were not statistically significant.

## 3. Discussion

The results of the present study indicate that nNOS is widely distributed in the nervous structures located in the duodenum of the domestic pig. This observation is in agreement with previous reports describing the presence of nitrergic neuronal cells and fibers in various fragments of the GI tract of several animal species, including humans [9,21,26,27,28]. On the other hand, it should be noted that knowledge about the distribution of nNOS in the ENS of the duodenum is rather scarce [9,14].

The majority of nitrergic neurons in the ENS belong to the class of nonadrenergic noncholinergic (NANC) neurons showing inhibitory activity [29]. NO in the GI tract mostly takes part in the relaxation of the intestinal muscles inhibiting the contractility of the smooth muscles in all segments of the GI tract from the esophagus to the rectum [19,30,31]. Moreover, NO is involved in the control of the secretion of electrolytes and intestinal hormones [22] as well as in the regulation of the mesenteric and intestinal blood flow through vasodilatory activity [21,26,32]. It should be noted that the role of this substance in the gastrointestinal secretion is not unequivocal. Although NO significantly reduces the gastric acid secretion [33], it increases the secretory activity within the duodenum and colon [24,34,35]. Previous studies have also shown that NO (strong inhibitory factor in the GI tract) is present in a small number of neurons synthesizing acetylcholine (the main stimulatory neuronal substance in the GI tract) [36]. The functions of such neurons, in which two neuronal factors have an opposite effect, are not clear. Some studies suggest that the abovementioned colocalization may occur in the myenteric descending interneurons taking part in the peristalsis [36].

Moreover, it has been noted that NO acts as a cytoprotective agent during some intestinal injuries and endotoxicities, mainly via the regulation of the mucosal blood flow [37,38]. It has also been reported that the endogenous release of this gaseous neurotransmitter induces protective effects of other active substances in the nervous system within the GI tract [39,40,41].

During the present study, the influence of T2 toxin on the number of nNOS-positive neurons has been observed in all parts of the ENS within the duodenum. As knowledge about the impact of T2 toxin on the ENS is fragmentary and NO in the enteric neurons can play multidirectional functions, the mechanisms underlying the observed changes are not clear. The observed fluctuations may result from various processes, including stimulation of the transcription and/or translation steps of the nNOS synthesis, as well as from perturbations in the transport of this molecule from perikaryon to presynaptic endings, where it participates in NO formation. Moreover, the increase in the number of nitrergic neurons may result from the abovementioned T2 toxin-induced stimulation together with cholinergic and nitrergic neurons to enhance their nNOS expression with the simultaneous inhibition of enzymes participating in acetylcholine synthesis. It is more likely that a decrease in the number of cholinergic neurons in the ENS has been reported under the impact of toxic substances [42].

The increase in the number of nNOS-LI enteric nervous structures observed, especially in the MP, may be connected with the anorectic activity of T2 toxin and its relaxant impact on the intestinal muscles [2,4,5]. Although no symptoms (such as lack of appetite or constipation) were observed during the experiment, changes associated with neuron activity may be the first signs of intoxication occurring before clinical symptoms.

The other reason for the observed changes may be connected with the relatively well known proinflammatory and damaging activity of T2 toxin, which is considered to be an important risk factor for leucocyte deficiency, Kashin–Beck disease, and alimentary toxic aleukia (ATA) [5,43,44]. This is very likely in view of the close interrelations between intestinal epithelial cells, the immunological system and the ENS, and the participation of NO in immune processes [45]. It has been previously reported that, depending on the inflamed tissue, NO may participate in both anti-inflammatory and proinflammatory reactions [46]. However, in the digestive tract, NO is primarily known as an important proinflammatory factor [47], which causes changes in the cytokine levels and takes part in processes associated with inflammatory bowel disease [14,48].

Another cause of the noted changes may be connected with the neurotoxic effects of T2 toxin [3]. Therefore, an increase in the number of nNOS-LI neurons noted in the present study may be associated with neuroprotective functions of NO. This thesis is supported by the fact that some previous studies have described the participation of NO in enhancing enteric neuron resistance to damaging factors [25] and the influence of nNOS on the expression of vital transcription factors in the enteric plexuses [49]. Nevertheless, it must not be forgotten that NO is also known as a strong damaging factor in the neuronal tissue [50] because, as a very active molecule, it may react in the cell body with many other substances leading to the blocking of important enzymes and contributing to the production of free radicals [50]. Such processes take place where the amount of NO in the tissues is high. So, it cannot be excluded that the increase in the percentage of nNOS-LI neurons noted in the present study is not connected with neuroprotective but, on the contrary, with neurodegenerative and neurotoxic processes induced by the T2 toxin.

Fluctuations in the number of nitrergic neurons under T2 intoxication may also result from changes in sensory and pain stimuli conduction. Some studies have described the presence of NO in the afferent nervous structures, including neurons supplying the GI tract [51], which may suggest the participation of NO in mechanisms involved in sensory processes. Although the doses of T2 used in the present study were low and no symptoms of the inflammatory processes and/or pain perception were observed during the study, it cannot be ruled out that an increase in the percentage of nNOS–LI neurons are the first sign of subclinical inflammatory processes and changes in sensory stimuli conduction.

It should also be noted that in the present study, the animals were treated with relatively low doses of T2 toxin. The doses used in the present study are lower than the lowest observed adverse effect level (LOAEL) for pigs established by the European Union. Therefore, the present study has shown that even low doses of T2 toxin may change the expression of nNOS in the ENS.

## 4. Materials and Methods

Ten immature female pigs of the Large White Polish breed (8 weeks old, 20 kg body weight (b.w.)) were used during the present study. The animals were obtained from a commercial fattening farm located in the vicinity of Olsztyn (Poland). After an adaptation period (five days), the animals were randomly divided into two groups of five gilts. One of them served as the control group (C group), and the second was the experimental group (T2 group). Each day, the animals were given capsules orally for the next 42 days. The capsules given to control animals were empty, while those given to the T2 group were filled with T2 toxin in a dose of 12 µg/kg b.w./day, which was calculated in relation to the current body weight of animals. This dose may be considered as a relatively low dose of T2 toxin, because it is clearly lower than the lowest observed adverse effect level (LOAEL) in pigs, established by the European Food Safety Authority at the level of 29 µg/kg b.w. per day [52].

Pigs were kept under standard laboratory conditions with access to food and water ad libidum as described previously by Makowska et al. [6], and all procedures were conducted in agreement with the directions of the Local Ethical Committee for Animal Experiments in Olsztyn (Poland) (decision from 28 November 2012, No. 73/2012/DTN). To avoid the influence of additional mycotoxin contamination, the feed used in the experiment was tested for the presence of these substances according to methods described previously by Makowska et al. [6] and contamination of feed with mycotoxins was not observed.

For the premedication and euthanasia of animals, which was made on the 43th day of the experiment, Stressnil (Janssen, Beerse, Belgium, 75 μL/kg b.w.) and an overdose of sodium thiopental (Thiopental, Sandoz, Kundl, Austria) were used, respectively. Tissues were collected from all pigs immediately after the death of the animal. Duodenal fragments (about 2 cm-long, 4–5 cm away from the pylorus) were fixed for one hour in a solution of 4% buffered paraformaldehyde (pH 7.4). These tissues were then transferred into the phosphate buffer (0.1 M, pH 7.4, at 4 °C) for three days with a daily exchange of this solution. On the fourth day, fragments of the duodenum were put into 18% phosphate-buffered sucrose, where they were stored for 3 weeks at 4 °C. The next step of tissue preparation was freezing at −22 °C in order to further cutting them into 14 µm-thick sections on microtome (Microm, HM 525, Walldorf, Germany). The cutting was performed perpendicular to the lumen of the duodenum.

The routine double-labeling immunofluorescence method used in this study was described previously by Gonkowski [17].

Briefly, this technique was performed as follows. Before labeling, the slides with tissue fragments were taken out of the freezer and dried for 45 min at room temperature (rt). For the next hour, they were incubated at rt with blocking solution (10% goat serum, 0.1% bovine serum albumin (BSA), 0.01% NaN3, Triton X-100, and thimerosal in phosphate-buffered saline - PBS). For the immunofluorescence labeling, duodenal fragments were incubated overnight (rt, humid chamber) with a mixture of two antibodies raised in different species directed toward: pan-neuronal marker protein gene-product 9.5 (PGP 9.5, mouse monoclonal antibody, catalogue No. 7863-2004, Biogenesis, UK, working dilution 1:2000) and a marker of nitrergic neuron neuronal isoform of nitric oxide synthase (nNOS, rabbit polyclonal antibody, catalogue no AB5380, MercMillipore, Darmstadt, DEU, working dilution 1:6000). The visualization of complexes of primary antisera bounded to appropriate antigen was made by the incubation (1 h, rt.) of tissue fragments with the mixture of two secondary antibodies conjugated with two types of Alexa Fluors: Alexa Fluor 488 (donkey antimouse IgG) and Alexa Fluor 546 (donkey antirabbit IgG). Both antibodies conjugated with Alexa Fluors were from Invitrogen (Carlsbad, CA, USA) and their working dilution was 1:1000. After each step of the immunofluorescence method, slides were rinsed with PBS (3 × 15 min, pH 7.4).

Labelled duodenum fragments were analyzed with an Olympus BX51 microscope with epi-fluorescence and appropriate filter sets connected with an Olympus XM10 camera.

For the exclusion of nonspecific labeling, three routine control tests of the antibodies were performed, i.e., preabsorption of the antiserum with the appropriate antigen, omission and replacement of primary antiserum by nonimmune serum.

To determine the influence of T2 toxin on nitrergic neuronal cells, the percentage of nNOS-like immunoreactive neurons in relation to all neuronal cells immunoreactive to protein gene product 9.5 (PGP 9.5—used a pan-neuronal marker) was evaluated. For this purpose, at least 500 PGP 9.5-positive perikaryons (only cells with clearly visible nuclei) in every type of duodenal enteric plexuses from each animal were examined for the presence of nNOS, and the number of PGP 9.5-LI cells was treated as 100%. The obtained results were pooled and presented as the mean percentage ± SEM. For cell counting, at least 10 fragments of the duodenum obtained from each animal spaced at least 150 µm apart were included in the study. This method was used to avoid double-counting the same perikaryons.

For the analysis of the number of nNOS-positive nerve fibers in the muscular and mucosal layers, all nerves immunoreactive to nNOS in the microscopic observation field (0.1 mm^2^) were counted. The counting of nerves was performed in 4 sections per animal with 5 fields per section. The obtained results were pooled and presented as the mean number of nerves ± SEM.

The average surface area of enetric ganglia (performed on 100 randomly selected ganglia from each type of the enteric plexus) was evaluated with ImageJ 7.1 (NIH open source software, Bethesda, MD, USA).

Student’s *t*-test was used for statistical analysis (Statistica 12, StatSoft, Inc., Cracow, Poland). Statistically significant differences were considered significant at p < 0.05.

## 5. Conclusions

In summary, the reason for the observed changes in the number of nNOS-LI nervous cells in the duodenal ENS under the impact of T2 toxin may be multidirectional, and the exact mechanisms of these processes need further studies. However, the present research has demonstrated that nitrergic neurons in the ENS participate in mechanisms connected with the influence of T2 toxin on the GI tract, and even low doses of this substance may impact a living organism. Therefore, exposure to even small amounts of this mycotoxin may be a threat to the health of both humans and animals.

## Figures and Tables

**Figure 1 ijms-21-05118-f001:**
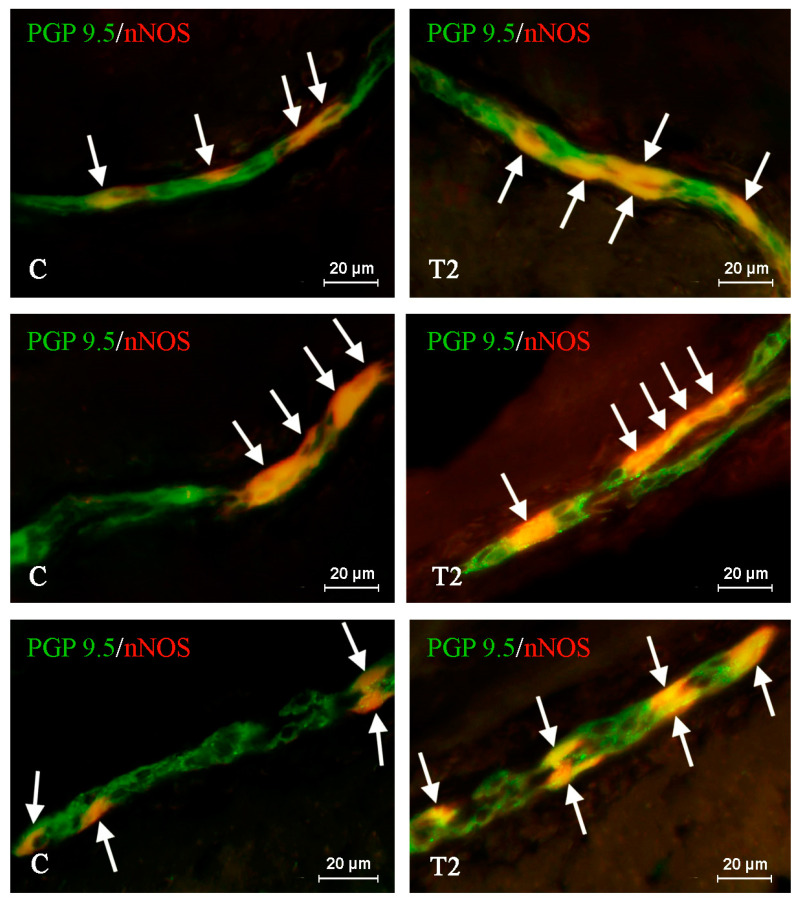
Distribution pattern of nerve cells immunoreactive to protein gene-product 9.5 (PGP 9.5)—used as pan neuronal marker and neuronal isoform of nitric oxide synthase (nNOS) in the myenteric plexus of porcine duodenum under physiological conditions (C) and after T2-toxin administration (T2); the pictures are the result of the overlap of both stainings. The arrows point to neurons immunoreactive for both PGP 9.5 and nNOS.

**Figure 2 ijms-21-05118-f002:**
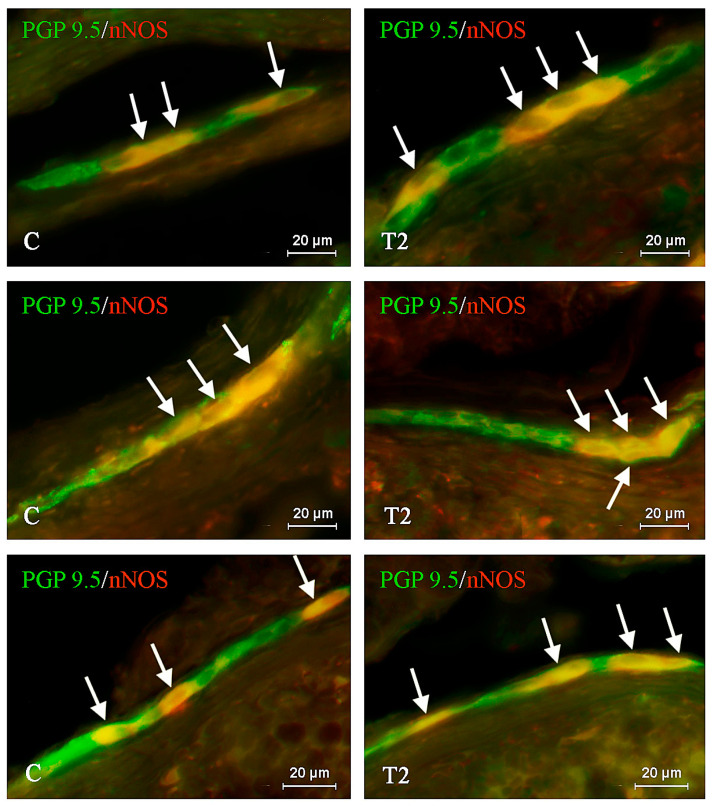
Distribution pattern of nerve cells immunoreactive to protein gene-product 9.5 (PGP 9.5)—used as pan neuronal marker and neuronal isoform of nitric oxide synthase (nNOS) in the outer submucous plexus of porcine duodenum under physiological conditions (C) and after T2-toxin administration (T2); the pictures are the result of the overlap of both stainings. The arrows are pointing neurons immunoreactive for both—PGP 9.5 and nNOS.

**Figure 3 ijms-21-05118-f003:**
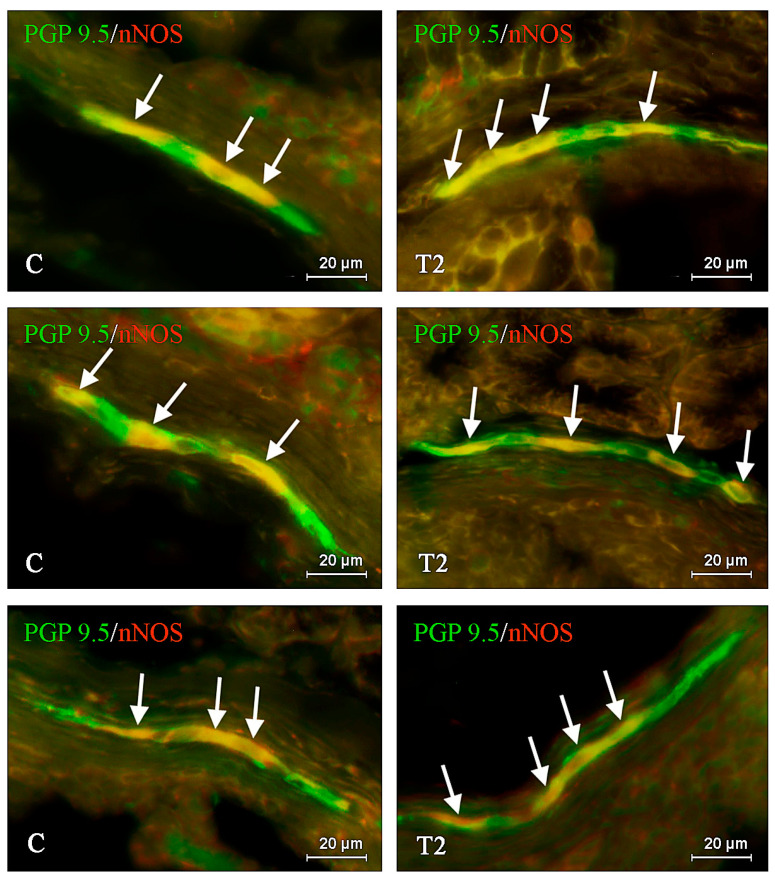
Distribution pattern of nerve cells immunoreactive to protein gene-product 9.5 (PGP 9.5)—used as pan neuronal marker and neuronal isoform of nitric oxide synthase (nNOS) in the inner submucous plexus of porcine duodenum under physiological conditions (C) and after T2-toxin administration (T2); the pictures are the result of the overlap of both stainings. The arrows are pointing neurons immunoreactive for both—PGP 9.5 and nNOS.

**Table 1 ijms-21-05118-t001:** Nitric oxide synthase (nNOS)-like immunoreactive perikarya and nerve fibers in porcine duodenum under physiological conditions (C group) and after administration of T2 toxin.

Part of the Intestinal Wall		C Group					T2 Group				
		Pig 1	Pig 2	Pig 3	Pig 4	Pig 5	Pig 1	Pig 2	Pig 3	Pig 4	Pig 5
**CML**	A	21.925	24.2	28.15	25.15	24.2	23.95	22.9	23.675	22.925	24.2
	average ± SEM	24.73 ± 1.008					23.53 ± 0.265				
**MP**	B	502/186	511/214	506/202	503/186	512/182	503/250	511/243	512/246	507/212	506/222
	C	37.05%	41.88%	39.92%	36.98%	35.55%	49.7%	47.55%	48.05%	41.81%	43.87%
	average ± SEM	38.28 ± 1.147% *					46.20 ± 1.453% *				
**OSP**	B	504/174	504/193	507/208	509/209	505/187	505/230	502/233	506/235	507/212	506/222
	C	34.52%	38.29%	41,03%	41,06%	37,03%	45,54%	46,41%	46,44%	44,05%	44,49%
	average ± SEM	38.39 ± 1.244% *					45.39 ± 0.488% *				
**ISP**	B	502/185	501/184	503/191	506/164	507/166	500/220	504/222	505/218	501/226	500/220
	C	36.85%	36.73%	37.97%	32.41%	32.74%	44.00%	44.05%	43.17%	45.11%	44.00%
	average ± SEM	35.34 ± 1.151% *					44.07 ± 0.308% *				
**ML**	A	7.45	9.675	11.225	10.25	9.675	8.3	9.825	9.675	8.75	9.225
	average ± SEM	9.66 ± 0.62					9.16 ± 0.285				

CML—circular muscle layer, MP—myenteric plexus, OSP—outer submucous plexus, ISP—inner submucous plexus, ML—mucosal layer, CB—cell bodies, NF—nerve fibers. Statistically significant (p ≤ 0.05) differences between C group and T2 group were marked with *. A—the average number of nNOS-positive nerve fibers per observation field (0.1 mm^2^) in the particular animals. B—the number of cells PGP 9.5+/nNOS+ counted in particular animals. C—the percentage of nNOS-positive cells in relation to the number of PGP 9.5—positive cells (treated as 100%).

**Table 2 ijms-21-05118-t002:** Size [µm^2^] of the MP—myenteric plexus, OSP—outer submucous plexus, and ISP—inner submucous plexus in porcine duodenum under physiological conditions (C group) and after administration of T2 toxin.

	C Group	T2 Group
MP	2619.13 ± 142.09	2524.08 ± 123.87
OSP	1879.69 ± 95.45	1634.08 ± 92.95
ISP	795.83 ± 46.2	695.81 ± 40.42

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
