# Peer review of "The T2 Toxin Produced by *Fusarium* spp. Impacts Porcine Duodenal Nitric Oxide Synthase (nNOS)-Positive Nervous Structures—The Preliminary Study"

_ijms, 2020, doi:10.3390/ijms21145118_

Round 1

Reviewer 1 Report

The paper by Rychlik et al assesses how a secondary metabolite of Fusarium species affect the autonomous nerve system of the gut in pigs. The study is descriptive and fails to assess any further study to highlight physiological relevance. That needs to be complemented.

The abstract must be re-written to reflect results.

Please use unbiased image analysis systems to evaluate histology.

What was the source of the animals used in the study?

The study up to that point is a descriptive study. Is there higher nitrative stress? Is there more NO produced?

Author Response

The authors thank the reviewer for evaluation of the manuscript, which allows to its improvement. Below are the answers to the Reviewer's 1 allegations.

1) The abstract must be re-written to reflect results.

The abstract has been re-written according to the suggestion of the Reviewer and now it shows the obtained results.

2) Please use unbiased image analysis systems to evaluate histology.

During the present study routine histopathological evaluation of the intestinal wall has been performed, but changes in the duodenal structures have not been observed. Therefore the authors decided to not put the photographs in the manuscript, because such photographs would show normal intestinal wall. Of course, if the Reviewer thinks it is appropriate the authors will add such photographs. The observation that intestinal villi under the impact of low doses of T2 are well developed and similar to those noted in the control group is in agreement with the previous studies on the influence of similar doses of T2 on the other part of the small intestines – on the jejunum (Obremski K, Zielonka L, Gajecka M, et al. Histological estimation of the small intestine wall after administration of feed containing deoxynivalenol, T-2 toxin and zearalenone in the pig. Pol J Vet Sci. 2008;11(4):339-345).

3) What was the source of the animals used in the study?

The source of animals has been added into the chapter “materials and methods”.

4) The study up to that point is a descriptive study. Is there higher nitrative stress? Is there more NO produced?

In the opinion of the authors, suggestion of the Reviewer that the evaluation of nitrative stress in the duodenum is interesting. Unfortunately the experiment has been completed and the authors are unable to do additional evaluations. The authors are aware that a lot of issues connected with the influence of T2 toxin on nitrergic innervation of the duodenum still remain unclear.

On the other hand, it should be pointed out that studies are of preliminary nature and the authors intend to continue more comprehensive study in this topic, but new experimental animals and a lot of time are needed to realize this aim. The preliminary character of the study has been underlined by the change of title of the article, which is now is now as follows: “The T2 toxin produces by Fusarium spp impacts porcine duodenal nitric oxide synthase (nNOS) – positive nervous structures – the preliminary study”.

The authors hope that explanations and manuscript improvements will allow to publish the article in the International Journal of Molecular Sciences.

Reviewer 2 Report

This is a potentially significant piece of work as it sheds light on common mycotoxic threats in our Food. However, there are several shortcomings which have to be addressed.

Major:

1) What is a "low dosis" in this context? What would be a high Dosis? Are there estimates how much of T2 might be contained in average food either for humans or animals?

2) There is insufficient information on morphometric procedures. How many sections 150 µm apart were used for neuronal cell counting? How many cell bodies (both PGP 9.5 and nNOS) were counted altogether?

3) The increase in nNOS positive neuronal cell bodies might be either a result of 1) a decrease of cholinergic Neurons or 2) an increase of nNOS expression in previously cholinergic-only neurons or 3) a de-novo proliferation of nitrergic neurons from enteric stem cells. From the data presented, this cannot be decided. Thus, as a first step, ChAT/nNOS double labeling should be performed in order to address at least issue #2. There is already normally a subpopulation of ChAT/nNOS co-positive enteric neurons in human and other species (Beck et al., Cell Tissue Res 2009, 338:37), and these neurons could be stimulated by T2 to enhance their nNOS expression. By the way, these possibilities should be considered at least in the Discussion which is much too speculative without additional data.

4) Instead of 14 µm sections, quantitative evaluation should be better performed on wholemounts. Thus, average ganglia size could be assessed. If the relative increase of nNOS neuron numbers is be the result of a decrease of cholinergic neurons, the average size of enteric ganglia should be smaller than in controls. This is much more difficult to study in sections.

5) As the number or Density of nNOS positive nerve fibers did not change, it may be concluded that also the number of nNOS perikarya did not change. This may be a hint to a possible decrease of cholinergic neuron numbers, and hence a relative increase of nNOS cell bodies. This should be another Motivation to perform ChAT or VAChT immunostaining in order to assess a possible decrase in cholinergic Fiber density.

6) In the micrographs provided, it is impossible to distinguish nerve fibers. Thus, the semiquantitative counting procedure cannot be evaluated.

Minor:

7) As references for general ENS anatomy (how many plexus in pig, etc.), the authors are encouraged to quote original Landmark papers by, e.g., Stach, Brehmer or Timmermans, many of which can be found in the Furness monograph on the ENS from 2006 (Blackwell).

Author Response

The authors thank for insightful review, which rises important and interesting issues and will allow to improve the manuscript.

Below are the answers to the Reviewer's 2 allegations.

1) What is a "low doses" in this context? What would be a high Doses? Are there estimates how much of T2 might be contained in average food either for humans or animals?

The term “low doses” of T2 toxin has been explained in the Materials and method by the phrase: “Capsules given to control animals were empty, while those given to the T-2 group were filled with T-2 toxin  in dose 12 µg/kg b.w./day, which was properly calculated in relation to current body weight of animals. This dose may be considered as relatively low dose of T2 toxin, because it is clearly lower than the lowest-observed-adverse-effect level (LOAEL) in pigs, established by European Food Safety Authority at the level of 29 µg/kg b.w. per day”.

2) There is insufficient information on morphometric procedures. How many sections 150 µm apart were used for neuronal cell counting? How many cell bodies (both PGP 9.5 and nNOS) were counted altogether?

Information about the number of sections included into the study has been added in “materials and methods”. Moreover, the authors reedited table in the chapter “results”. Now, the table presents exact number of cells immunoreactive to panneuronal marker PGP 9.5 and nNOS counted in the particular animals during the study.

3) The increase in nNOS positive neuronal cell bodies might be either a result of 1) a decrease of cholinergic neurons or 2) an increase of nNOS expression in previously cholinergic-only neurons or 3) a de-novo proliferation of nitrergic neurons from enteric stem cells. From the data presented, this cannot be decided. Thus, as a first step, ChAT/nNOS double labeling should be performed in order to address at least issue #2. There is already normally a subpopulation of ChAT/nNOS co-positive enteric neurons in human and other species (Beck et al., Cell Tissue Res 2009, 338:37), and these neurons could be stimulated by T2 to enhance their nNOS expression. By the way, these possibilities should be considered at least in the Discussion which is much too speculative without additional data.

It is an interesting suggestions. Unfortunately, currently the authors do not have antibodies against VAChT and/or ChAT. Of course the authors may buy the antibodies, but tender procedures at the university usually last about 3 months (now they can be longer due to pandemic covid-19). Moreover, the authors know that a lot of aspects connected with the influence of T2 toxin on nitrergic innervation are not clear (for example co-localization of nNOS with other substances, reaction of the extrinsic innervation on T2 toxin administration and many others) and the present manuscript is de facto the preliminary study. This fact is underlined by the change in the manuscript title, which now is: “The T2 toxin produces by Fusarium spp impacts porcine duodenal nitric oxide synthase (nNOS) – positive nervous structures – the preliminary study”. Simultaneously the evaluation of the surface area of the enteric ganglia has been additionally made (see the answer for point 5). The obtained results show that T2 toxin in this experiment did not caused changes in the surface of enteric ganglia, what may suggest that the number of cholinergic neurons did not change.  Moreover to three points mentioned by the Reviewer as a reason of the increase in the number of nNOS-positive cells, the authors would add the changes (block) in the transport of nNOS from cell body to nerve endings and the increase in the intensity of nNOS production. It should be remembered that immunofluorescence method show neurons at a given moment. Neurons, even capable of production of nNOS, but not producing it at the moment or transporting the whole amount of this substance to nerve endings are visible as negative cells. According to the population of ChAT/nNOS it is very interesting population, which certainly merits further investigation, but currently the authors cannot do such investigation. However, the description of possibilities of connections between the increase in the number of nNOS-positive neurons with decrease in the number of cholinergic neurons has been added to the discussion chapter. Moreover discussion has been shortened and the authors hope that now it is not much too speculative.

4) Instead of 14 µm sections, quantitative evaluation should be better performed on wholemounts. Thus, average ganglia size could be assessed. If the relative increase of nNOS neuron numbers is the result of a decrease of cholinergic neurons, the average size of enteric ganglia should be smaller than in controls. This is much more difficult to study in sections.

Of course the authors are in agreement that wholemounts technique is better from sections labelling. But on the other hand immunofluorescence technique on sections is also accepted in such type of the study. Moreover, according the suggestions of the Reviewer the study on the influence of T2 on the average surface area of enteric ganglia has been performed using ImageJ 7.1 (NIH open source software, USA). The obtained results show that there are no statistically significant differences between control animals and pigs receiving T2 toxin. According to suggestions of the Reviewer such results suggest that the increase in the number of nNOS-positive neurons are not connected with the decrease in the number of cholinergic cells.

5) As the number or Density of nNOS positive nerve fibers did not change, it may be concluded that also the number of nNOS perikarya did not change. This may be a hint to a possible decrease of cholinergic neuron numbers, and hence a relative increase of nNOS cell bodies. This should be another Motivation to perform ChAT or VAChT immunostaining in order to assess a possible decrease in cholinergic Fiber density.

It is a valid comment, but as mentioned above unfortunately the authors don't have antibodies (against ChAT or VACht) right now. Moreover, it should be pointed out that nerves located in the mucosal and muscular layers of the duodenum are not only processes of the enteric neurons located in the enteric ganglia. Nerves may be also the processes of neurons located outside the intestine (extrinsic innervation of the GI tract), which may be located in sympathetic and parasympathetic ganglia, as well as in dorsal root ganglia. At present we do not know how extrinsic innervation of the intestine reacts on the administration of T2 toxin. Moreover, the increase in the number of nNOS-positive cells (what suggest the increase in nitric oxide synthesis) may be accompanied by blockage of transport of nNOS to nerve endings. In this case the increase in the number of nNOS-positive cells may be connected with compensatory increase in NO synthesis being the answer to the blocking of nNOS transport to nerve ending and low doses of NO in synapses. So, in opinion of the authors the lack changes in the number of nerve fibers is not basis to conclude that the number of nNOS perikarya also does not change.

6) In the micrographs provided, it is impossible to distinguish nerve fibers. Thus, the semiquantitative counting procedure cannot be evaluated.

The Reviewer is right that on figures selected to publication intraganglionic nerves are not visible. This is because the main subject of the manuscript were nNOS-positive cells, and the authors selected photographs with good visible neurons. Intraganglionic nerves immunoreactive to nNOS in the majority of ganglia were rather weak visible. The authors have completely reedited the tables with results and (due to the fact that semiquantitative evaluation is rather not accurate and according to suggestion of the Reviewer such procedure cannot be evaluated) decided to remove the part of results which relate to this issue.

Minor:

7) As references for general ENS anatomy (how many plexus in pig, etc.), the authors are encouraged to quote original Landmark papers by, e.g., Stach, Brehmer or Timmermans, many of which can be found in the Furness monograph on the ENS from 2006 (Blackwell).

The references has been changed according to the suggestion of the Reviewer.

The authors hope that explanations and manuscript improvements are in line with the wishes of the Reviewer and will allow to publish the manuscript in International Journal of Molecular Science.

Round 2

Reviewer 1 Report

The authors have not really improved their paper, hence, it is of low quality to be published.

Reviewer 2 Report

Thank you for this extensive revision! I have no further criticism.